# OpenReview forum: "Beyond Levels and Continuity: A New Statistical Method for DNNs Robustness Evaluation"
_ICLR.cc/2025/Conference — ICLR 2025 Conference Withdrawn Submission_

### Official Review · Reviewer_EowM · 2024-10-29

**Soundness:** 3
**Presentation:** 3
**Contribution:** 2
**Rating:** 5
**Confidence:** 3

**Summary:**

This work addresses the challenge of evaluating adversarial robustness in deep neural networks (DNNs), particularly for large models where traditional white-box verification methods are infeasible. The authors introduce a novel statistical framework, REPP, which redefines the probability of encountering adversarial examples as a Poisson-based problem, enabling the development of a Minimum Variance Unbiased Estimator (MVUE) that improves estimation accuracy. Unlike previous approaches, REPP does not require continuity or level conditions, making it broadly applicable and effective across domains like computer vision and natural language processing, with demonstrated success on large models including ViT classifiers and LLMs.

**Strengths:**

1. The paper introduces REPP, a unique statistical framework that redefines the probability of adversarial examples using a Poisson-based approach, allowing for more precise and unbiased estimation of adversarial risk.

2. Unlike traditional methods that rely on continuity or level-based conditions, REPP is adaptable to both continuous and discontinuous outputs, making it broadly applicable across diverse domains, including natural language processing and computer vision.

3. REPP reduces the number of queries and simulations required.

**Weaknesses:**

1. The motivation behind the proposed method is unclear. If a service provider wants to evaluate their model's robustness, they could simply use white-box evaluation methods. Conversely, if a user with only black-box access to the model API attempts to assess robustness, they would need to make millions of queries, potentially incurring significant costs.

2. The experimental evaluation is limited. The experiments are conducted only on the MNIST dataset and 200 ImageNet images, and the single baseline used is outdated (published in 2018). To better demonstrate the proposed method's effectiveness, the authors should include a broader range of datasets and comparison baselines, including both black-box adversarial attacks and basic white-box attacks, such as FGSM.

**Questions:**

See weaknesses

---

> ### Author Response · Authors · 2024-11-28
> **Response to reviewer EowM**
>
> We truly appreciate the great comments from the reviewer, and here are our responses to the proposed questions. If you find our response satisfactory, please consider raising your score.
>
> **W1** The motivation behind the proposed method is unclear. If a service provider wants to evaluate their model's robustness, they could simply use white-box evaluation methods. Conversely, if a user with only black-box access to the model API attempts to assess robustness, they would need to make millions of queries, potentially incurring significant costs.
>
> > It is true that white-box evaluation methods would be easier for evaluating the robustness. However, this also depends on the adversary's knowledge, note that REPP only requires querying the model in a black-box manner.
> In practice, we may not be able to access the model weight/structure and perform white-box evaluation.
> This is why there are plenty of approaches proposed for black-box robustness evaluation in a deterministic or statistical manner [1-10].
>
> [1] A statistical approach to assessing neural network robustness, ICLR 2018
>
> [2] Certified adversarial robustness via randomized smoothing, ICML 2019
>
> [3] Black-box certification with randomized smoothing: A functional optimization based framework, 2020
>
> [4] Efficient statistical assessment of neural network corruption robustness, ICLR 2021
>
> [5] Scalable quantitative verification for deep neural networks, ICSE 2021
>
> [6] CC-Cert: A probabilistic approach to certify general robustness of neural networks, AAAI 2022
>
> [7] Statistical Certification of Acceptable Robustness for Neural Networks, ICANN 2021
>
> [8] RoMA: A Method for Neural Network Robustness Measurement and Assessment, NIP 2022
>
> [9] Proa: A probabilistic robustness assessment against functional perturbations ECML 2022
>
> [10] Towards verifying the geometric robustness of large-scale neural networks, AAAI 2023
>
> **W2** The experimental evaluation is limited. The experiments are conducted only on the MNIST dataset and 200 ImageNet images, and the single baseline used is outdated (published in 2018). To better demonstrate the proposed method's effectiveness, the authors should include a broader range of datasets and comparison baselines, including both black-box adversarial attacks and basic white-box attacks, such as FGSM.
>
> > We thank the reviewer for the comment. But note that we were not only conducting experiments on the Image dataset but also in the NLP domain.
>  We agree that a broader range of datasets may further strengthen the justification, but regarding the adversarial attacks, in the ImageNet experiment, we directly compared REPP with both the deterministic verification and statistical verification methods.
> Verification is a more challenging task than adversarial attacks, as adversarial attacks can only provide the upper-bound performance without any guarantee.
> Although the proposed REPP produces adversarial examples during estimation, this is a byproduct, not the main purpose of the method (it tells how rare we may encounter adversarial examples statistically).

---

### Official Review · Reviewer_VVLd · 2024-10-30

**Soundness:** 2
**Presentation:** 1
**Contribution:** 3
**Rating:** 3
**Confidence:** 3

**Summary:**

The paper proposes a statistical method to assess the likelihood of an adversarial example existing in a given perturbation set. In particular, the problem is formulated as a rare-event estimation problem and, compared to previous work, allows for discrete input domains. The method is evaluated for continuous perturbation sets e.g., in the image domain and for discrete perturbation models in the NLP domain.

**Strengths:**

* Understanding the likelihood of adversarial examples occurring is an interesting problem and considering how to extend existing work to deal with discrete input domains is a natural continuation
* The poison process modeling framework used seems to be a valid approach to solve the problem
* The method is applicable to semantically more interesting perturbation sets such as e.g., geometric transformations for images or adversarial suffixes for LLMs

**Weaknesses:**

1. The main claim of the paper does not match the contribution. In particular, the paper claims to develop a minimum variance unbiased estimator (MVUE) to accurately estimate the likelihood of encountering adversarial examples. However, the paper "only" uses the MVUE developed by Walter (2015), which does not require any adaptation in the derivation and can be applied to arbitrary random variables. The novelty is applying the existing estimator to a random variable describing whether there exists an adversarial example. In particular nearly the complete Methods Section 4, i.e., Definitions 1, 2, 3, 4, Theorems 1, 2, 3, Propositions 1 are (close to) direct copies from Walter (2015) or standard results as mentioned by Walter (2015), but this is only indicated for Definition 3 and Theorem 3. While the authors do cite Walter (2015) before Section 4, it was not clear to me while reading that these results are already existing.
2. The focus of presentation in the paper seems odd. I do think it is very valid to explore an existing statistical estimator for the problem studied in the paper, however, in the presentation of the method (i.e., the methods section), I would expect a description of the statistical estimator and how/which adaptions are necessary for the particular ML application as done e.g. by Webb et al. (2019) regarding AMLS, and not a derivation of the (existing) statistical method itself in a theorem-proof style (without making clear that these are standard/existing results as mentioned above). E.g., the complexity discussion, pseudocode, and description how to actually leverage the method to estimate the probability for adversarial examples is moved to the appendix and only briefly mentioned at the beginning of Section 5.
3. The provided experimental details seem limited for reproducibility.
   * No code is attached to the submission.
   * There is no reproducibility statement.
   * F.1 and F.3 just state "We mainly follow the same settings in the AMLS paper (Webb et al. 2018).". I would expect to state concretely if one follows the same settings and if only mainly, state where one deviates. Also, there is an "as" missing.
   * I think providing more details and/or pseudocode for the employed Gipps sampling (5.2.1.) or Metropolis Hasting (5.1) would help reproducibility. Mention the MCML sampling used in 5.2.2.
   * I'm in general missing details how sampling is performed from the perturbation sets (i.e., Line 1 in Algorithm 1)
4. There seem to be small technical errors and inaccuracies in language/definitions.
   * The arrival times $(T_n)_{n\ge 0}$ of the Poisson process are claimed to be a homogeneous Poisson process with parameter 1. Indeed, the proof only shows that the inter-arrival times follow an exponential distribution leading to the counting process being a Poisson process. The arrival times of a Poisson process are continuous and distributed like the Gamma distribution.
   * I'm confused about the hypothesis test setting. The hypothesis seem not clearly defined, but the text rather defines the terms *robustness violation* and *probabilistically certified robust*.
   * Also, in $\mathcal{H}_1$ it states "[...] and the estimated probability $\hat{\mathcal{I}}$ [...] satisfying [...]", but a condition on $\hat{\mathcal{I}}$ is never stated and the probability statement regarding the confidence only holds for $N \rightarrow \infty$ not achievable in practice.
   * In the main body, it is never stated that $(T_n)_{n\ge 0}$ are the arrival times, which was confusing to me among others, as the distribution was claimed to be Poisson.
   * Theorem 2: $M_y$ follows a Poisson law with parameter $t_y = - \log P[Y \ge y]$ and not $t_y = - \log \mathcal{I}$
   * Theorem 3: "strict inequality random walks" are never introduced in the paper, thus reading "non-strict inequality random walks" feels confusing.
   * Provide domain definitions for input / labels / outputs at the beginning of Section 3.
5. In general, the paper together with the points above, feels unpolished.
   * In the beginning of Section 5 the authors make an argument using the variance of the estimator. But at this point of the paper, the variance of the estimator is not discussed, only later in Line 376 there is a short reference to the appendix discussing the variance.
   * There are some sentences that feel "lost" e.g., Line 122/23 ", where a base classifier ... [...]."
   * Appendix A regarding background knowledge on the Poisson Process is never cited. It would have helped me to reference it at the start of Section 4.
   * There are many grammatical issues and I recommend running a spell checker such as Grammarly.
6. The Conclusion is just a summary of the paper, I'm missing a discussion on potential limitations, open problems, and potential ways forward.

**Questions:**

1. GeoRobust is claimed to be complete given sufficient computational resources. Are the results in Table 1 using GeoRobust referring to complete or incomplete certification? In line 421/22 it is claimed that REPP outperforms the optimal result from GeoRobust. But if GeoRobust's optimal result is a complete certificate, this means that REPP achieving higher certified robustness is evidence of the unsoundness of REPP.
2. Related, is it correct that REPP cannot provide a sound certificate? I would expect a discussion on the soundness as this can be a limitation compared to deterministic verification approaches or probabilistic approaches such as randomized smoothing.
3. I'm excited about the applicability to more complex adversarial perturbation sets. E.g., it would be interesting to certify against common corruptions (Hendrycks et al. 2019) as included e.g. in the Robust Bench benchmark. Have I understood correctly, that the only requirement is that we have to be able to sample i.i.d. from the perturbation set?
4. For me, it is not 100% clear how previous work, e.g. by Webb et al. (2019) is not applicable to more complex adversarial perturbation sets. Taking not a fixed translation, but e.g. an arbitrary translation sampled between no translation and a translation up to a certain point. Would then AMLS still be applicable as we include all translations up to a certain point, making the input & output distribution continous again? I think a general discussion on the applicability to more complex (arbitrary) perturbation models and how previous work is limited in that respect would be helpful.
5. I'm missing the parameter $N$ for REPP in Section 5.2.2. Also, can you explain the context that you arrive at 25000 tokens?
6. The estimated log probability in Section 5.2.2 are many orders of magnitutde below the permissible probability level for the upper bound. Thus, I wonder how large is the upper bound compared to the estimate? What does it mean if the upper bound is so loose? Further, I wonder, how telling the resulting log-probability really is (especially if so small as reported in Table 2). Here, the perturbation space is tremendously huge and what matters for reliability is probably how easy it is for an adversary to reach a subset in the perturbation space through optimization or some other guiding principle, rather than the size of the adversarial subset per se.
7. Given the confidence interval is only approximate. How valid is it/how confident can we be for finite $N$? Or even small $N$ as proposed in Section 5 for verification?
8. What is the computational complexity of Algorithm 1 and what are the practical runtimes for the algorithm in the reported experiments in the case that either an adversarial example exists, or in the case that none exists.
9. Could you elaborate a bit more on the argument why for verification a smaller $N$ suffices?
10. Why does the number of calls for estimating the probability $\mathcal{I}$ scale with $\mathcal{O}(\log(1/\mathcal{I}))$?

Minor things:
* Line 174: The concept of a level is not introduced, but plays an important role in discussing the related work and contribution.
* Line 312: It reads as if $\hat{\mathcal{I}}$ has no effect on the discontinous case, but the discontinous case has no effect on  $\hat{\mathcal{I}}$
* Line 312: Second sentence writes $\hat{\mathcal{I}}$ [...] special version of $\hat{\mathcal{I}}$"
* Line 520: 10 and not 6 behaviors

**References**
Walter, "Rare event simulation and splitting for discontinuous random variables", ESAIM: Probability and Statistics 2015
Webb et al., "A Statistical Approach to Assessing Neural Network Robustness", ICLR 2019
Hendrycks & Dietterich, "Benchmarking Neural Network Robustness to Common Corruptions and Perturbations", ICLR 2019

---

> ### Author Response · Authors · 2024-11-28
> **Response to reviewer VVLd - part 1**
>
> We truly appreciate the great comments from the reviewer, and here are our responses to the proposed questions. If you find our response satisfactory, please consider raising your score.
>
> **W1**: The main claim of the paper does not match the contribution. In particular, the paper claims to develop a minimum variance unbiased estimator (MVUE) to accurately estimate the likelihood of encountering adversarial examples. However, the paper "only" uses the MVUE developed by Walter (2015), which does not require any adaptation in the derivation and can be applied to arbitrary random variables. The novelty is applying the existing estimator to a random variable describing whether there exists an adversarial example. In particular nearly the complete Methods Section 4, i.e., Definitions 1, 2, 3, 4, Theorems 1, 2, 3, Propositions 1 are (close to) direct copies from Walter (2015) or standard results as mentioned by Walter (2015), but this is only indicated for Definition 3 and Theorem 3. While the authors do cite Walter (2015) before Section 4, it was not clear to me while reading that these results are already existing.
>
> > We thank the reviewer point this out. To avoid overclaiming, we have removed the word 'develop' and merely used 'adapt' to avoid misleading.
> It is noted that we have cited (Walter, 2015) at the beginning. We also thank the reviewer agreed that the novelty is applying the existing estimator to a random variable describing whether there exists an adversarial example. Although the point process is generally applied to arbitrary random variables, it is not easy to adapt it to the case of calculating the probability of adversarial examples.
>
> **W2**: The focus of presentation in the paper seems odd. I do think it is very valid to explore an existing statistical estimator for the problem studied in the paper, however, in the presentation of the method (i.e., the methods section), I would expect a description of the statistical estimator and how/which adaptions are necessary for the particular ML application as done e.g. by Webb et al. (2019) regarding AMLS, and not a derivation of the (existing) statistical method itself in a theorem-proof style (without making clear that these are standard/existing results as mentioned above). E.g., the complexity discussion, pseudocode, and description how to actually leverage the method to estimate the probability for adversarial examples is moved to the appendix and only briefly mentioned at the beginning of Section 5.
>
> > We thank the reviewer for pointing out the problem with the current presentation, which is due to the space limit. We will reorganize the content, remove the unnecessary proof, and directly cite Walter (2015) for the lemma. Our initial intention was to help readers understand the entire method, which is why we chose to reintroduce certain definitions and theorems where we felt it was necessary.
>
> **W3.1**: No code is attached to the submission.
>
> > The project code will be released upon acceptance.
>
> **W3.2**: There is no reproducibility statement.
>
> > According to the [AuthorGuide](https://iclr.cc/Conferences/2025/AuthorGuide) in ICLR2025, the reproducibility statement is encouraged/optional but not compulsory.
>
> **W3.3**: F.1 and F.3 just state "We mainly follow the same settings in the AMLS paper (Webb et al. 2018).". I would expect to state concretely if one follows the same settings and if only mainly, state where one deviates. Also, there is an "as" missing.
>
> > We thank the reviewer for pointing this out. As we have reproduced all the original experiments from them regarding these two datasets as baselines and added our results merely. We have stated concretely with more details.
>
> **W3.4**: I think providing more details and/or pseudocode for the employed Gipps sampling (5.2.1.) or Metropolis Hasting (5.1) would help reproducibility. Mention the MCML sampling used in 5.2.2.
>
> > As suggested, we have added the pseudo-code for both methods in Appendix B.2 and B.3 (colored in blue)
>
> **W3.5**: I'm in general missing details how sampling is performed from the perturbation sets (i.e., Line 1 in Algorithm 1)
>
> > A simple example can be understood as: given the input $x_0$, and $x = x_0+\delta, s.t. ||\delta\||_\infty = \epsilon$, then the potential perturbation sets can be sampled uniformly as $x\sim \mathcal{U}(x_0-\epsilon,x_0+\epsilon)$

---

> ### Author Response · Authors · 2024-11-28
> **Response to reviewer VVLd - part 2**
>
> **W4.1**: The arrival times $(T_n)_{n\ge 0}$ of the Poisson process are claimed to be a homogeneous Poisson process with parameter 1. Indeed, the proof only shows that the inter-arrival times follow an exponential distribution leading to the counting process being a Poisson process. The arrival times of a Poisson process are continuous and distributed like the Gamma distribution.
>
> > Thank you for pointing out this important question.
> Indeed, since the inter-arrival times are exponentially distributed, the arrival times $(T_n)_{n \geq 1}$ are cumulative sums of independent exponential random variables, the inter-arrival times being independent and exponentially distributed is sufficient to establish that the arrival times form a [homogeneous Poisson process](https://www.itl.nist.gov/div898/handbook/apr/section1/apr171.htm).
>
> > The Gamma distribution also describes the statistical property of arrival times, while the Poisson process definition ensures the independence and homogeneity of the process. The arrival times $(T_n)_{n \geq 1}$ satisfy both the Gamma distribution and the properties of a homogeneous Poisson process. The two perspectives are complementary rather than contradictory.
>
> **W4.2**: I'm confused about the hypothesis test setting. The hypothesis seem not clearly defined, but the text rather defines the terms robustness violation and probabilistically certified robust.
>
> > The hypothesis test setting is defined and described in lines 151-158. Please let us know which parts confused you.
>
> **W4.3**: Also, in $\mathcal{H}_1$ it states "[...] and the estimated probability $\hat{\mathcal{L}}$  [...] satisfying [...]", but a condition on $\hat{\mathcal{L}}$ is never stated and the probability statement regarding the confidence only holds for $N\rightarrow \infty$
>
> > $\hat{\mathcal{L}}$ is used in Equation (8) as the estimating probability, we have stated it more clearly. And most hypothesis testing with intervals is indeed an asymptotic approximation.
>
> **W4.4**: In the main body, it is never stated that $(T_n)_{n\ge 0}$ are the arrival times, which was confusing to me among others, as the distribution was claimed to be Poisson.
>
> > It is true that we never stated that $(T_n)_{n\ge 0}$ are the arrival times, instead we describe it as the associated time sequence to avoid 'arrival' being misunderstood as the time of the adversarial example occurs. We have updated the content in the main body as suggested.
>
> **W4.5**: Theorem 2: $M_y$ follows a Poisson law with parameter with parameter $t_{y} = -\log P[Y>y]$ and not $t_{y} = -\log \mathcal{I}$
>
> > We admit that here should be $t_{y} = -\log P[Y>y]$ and $\mathcal{I}$ is a special case when $y=0$ and $\mathcal{I}=P[Y>y]$ defined in line 161.
>
> **W4.6**: Theorem 3: "strict inequality random walks" are never introduced in the paper, thus reading "non-strict inequality random walks" feels confusing.
>
> > The difference between strict inequality and non-strict inequality has been introduced in Definition 2, it represents the target distribution during the random walk $P(\cdot|Y>d)$ and $P(\cdot|Y\ge d)$, we will state clearer.
>
> **W4.7**: Provide domain definitions for input / labels / outputs at the beginning of Section 3.
>
> > As our method can be applied to different domains, CV/NLP, we wrote it in a more general way, but we will update it accordingly.
>
> **W5.1**: In the beginning of Section 5 the authors make an argument using the variance of the estimator. But at this point of the paper, the variance of the estimator is not discussed, only later in Line 376 there is a short reference to the appendix discussing the variance.
>
> > As the argument has been made, and acturally in Fig. 1 also plots the variance as the error bars, but they are barely visible due to both of small variances.
>
> **W5.2**: There are some sentences that feel "lost" e.g., Line 122/23 ", where a base classifier ... [...]."
>
> > The base classifier here is the original classifier used in randomized smoothing. We have polished the description to make it clear.
>
> **W5.3**: Appendix A regarding background knowledge on the Poisson Process is never cited. It would have helped me to reference it at the start of Section 4.
>
> > We have referred to Appendix A at the beginning of Section 4 as suggested.
>
> **W5.4**: There are many grammatical issues and I recommend running a spell checker such as Grammarly.
>
> > We will do that as suggested.
>
> **W6**: The Conclusion is just a summary of the paper, I'm missing a discussion on potential limitations, open problems, and potential ways forward.
>
> > Due to the space limit, we now have added a discussion in Appendix K.

---

> ### Author Response · Authors · 2024-11-28
> **Response to reviewer VVLd - part 3**
>
> **Q1**: GeoRobust is claimed to be complete given sufficient computational resources. Are the results in Table 1 using GeoRobust referring to complete or incomplete certification? In line 421/22 it is claimed that REPP outperforms the optimal result from GeoRobust. But if GeoRobust's optimal result is a complete certificate, this means that REPP achieving higher certified robustness is evidence of the unsoundness of REPP.
>
> > It is true that GeoRobust is claimed to be complete given sufficient computational resources, but given in practice they only run for limited iteration, making it an incomplete approach.
>
> **Q2**: Related, is it correct that REPP cannot provide a sound certificate? I would expect a discussion on the soundness as this can be a limitation compared to deterministic verification approaches or probabilistic approaches such as randomized smoothing.
>
> > We would never claim that REPP provides a sound certificate, as we terminate the algorithm when the estimating probability is lower than the pre-defined threshold.
> Also, we do not aim to claim a statistical approach can beat deterministic verification, instead, it can provide more information and some confidence in the model deployed when deterministic methods are infeasible.
> Although the method we proposed falls into statistical verification, the definition are entirely different from Randomized Smoothing. It is noted that Randomized Smoothing can be seen as a defense approach, and it aims to construct a smoothed classifier that is robust to input perturbations and provides certified adversarial robustness on the smoothed classifier. However, our method, REPP directly estimates the probability of encountering adversarial examples on the base classifier, therefore they can not be compared directly in this case. One possible comparison we can see is to compute the probability of the adversarial example of the smoother classifier given the certified radius by RS, to see whether both probabilities match. However, this may be a bit out of the scope of this work, but we are willing to put this comparison in the future study.
>
>
> **Q3**: I'm excited about the applicability to more complex adversarial perturbation sets. E.g., it would be interesting to certify against common corruptions (Hendrycks et al. 2019) as included e.g. in the Robust Bench benchmark. Have I understood correctly, that the only requirement is that we have to be able to sample i.i.d. from the perturbation set?
>
> > We appreciate that the reviewer pointed out the common corruptions will strengthen the experiment, given the limited time during the rebuttal period, we will add it in the next updated version.
> We would also like to argue that, actually the sample i.i.d. from the perturbation set is not what we need, instead we can perform a random walk in the Markov chain, we would refer to the above pseudo-code for better understanding.
> On the contrary, those methods relying on the concentration inequality bound get a probability estimation, like Chernoff (Baluta et al., 2021), Chernoff-Cramer (Pautov et al., 2022), Hoeffding (Huang et al., 2021), and Adaptive Hoeffding (Zhang et al., 2022),  their requirement is able to sample i.i.d. from the perturbation set. However, as stated in lines 114-120, these methods will fail to provide meaningful results, which is also one of the contributions of our work.
>
> **Q4**: For me, it is not 100% clear how previous work, e.g. by Webb et al. (2019) is not applicable to more complex adversarial perturbation sets. Taking not a fixed translation, but e.g. an arbitrary translation sampled between no translation and a translation up to a certain point. Would then AMLS still be applicable as we include all translations up to a certain point, making the input & output distribution continuous again? I think a general discussion on the applicability to more complex (arbitrary) perturbation models and how previous work is limited in that respect would be helpful.
>
> > It is true that AMLS is supposed to be able to be applied in this geometric experiment, it can still be a continuous case thus fall into Section 5.1. However, the main purpose of this experiment is not to show that we can compute the adversarial density (which would be similar to the MNIST experiment), here we would like to reveal the issue in the previous statistic methods relying on independent sampling along with the concentration inequality bound would lose their effectiveness in this case.
>
> **Q5**: I'm missing the parameter N or REPP in Section 5.2.2. Also, can you explain the context that you arrive at 25000 tokens?
>
> > The number of tokens here refers to the vocabulary used in llama2 (originally 32000, and about 25000 excluding the ASCII values.

---

> ### Author Response · Authors · 2024-11-28
> **Response to reviewer VVLd - part 4**
>
> **Q6**: The estimated log probability in Section 5.2.2 are many orders of magnitutde below the permissible probability level for the upper bound. Thus, I wonder how large is the upper bound compared to the estimate? What does it mean if the upper bound is so loose? Further, I wonder, how telling the resulting log-probability really is (especially if so small as reported in Table 2). Here, the perturbation space is tremendously huge and what matters for reliability is probably how easy it is for an adversary to reach a subset in the perturbation space through optimization or some other guiding principle, rather than the size of the adversarial subset per see.
>
> > Given $\alpha = 1e-15$ and N=20, the largest estimated log probability (−51.37) gives an upper bound of (-38.20), and the largest estimated log probability (−51.37), for the smallest estimated log probability in our cases (< −135.78), it gives an upper bound of -115.40 ($\mathcal{I}<1e-50$).
> This means the upper bound is not loose.
> Additionally, this example further validates why currently LLMs are popular in people's daily lives, where the low log probabilities of adversarial occurrence illustrate the hardness of a normal user to induce the harmful content from the LLM with a prompt with a random suffix.
> However, for researchers in the adversarial community, a low probability does not represent the non-existence of an adversary, where they can still be induced with advanced adversarial strategy.
> However, it also depends on the adversary's knowledge, note that REPP only requires querying the model in a black-box manner.
>
> **Q7**: Given the confidence interval is only approximate. How valid is it/how confident can we be for finite $N$? Or even small $N$ as proposed in Section 5 for verification?
>
> > It is True that even for small $N$, we fall into the range of validity of this approximation, as long as the $\mathcal{L}$ is small and certain $N$: for instance, considering $N \ge 10$ and $\mathcal{I} \le 0.1$ leads to $−N \log \mathcal{I}$ ≥ 23, in statistics, when the Poisson rate parameter $\lambda \geq 10$, the Poisson distribution can be well approximated by a Gaussian (normal) random variable. It is based on empirical evidence that, for most real-world distributions, the central limit theorem ensures the sampling distribution is approximately normal at this sample size. While not a strict cutoff, it provides a practical guideline for when the normal approximation is sufficiently accurate for hypothesis testing and confidence intervals. In our case, the sample size and underlying assumptions make this approximation valid.
>
>
> **Q8**: What is the computational complexity of Algorithm 1 and what are the practical runtimes for the algorithm in the reported experiments in the case that either an adversarial example exists, or in the case that none exists.
>
> > Fig.2 shows the comparison of computation time.
> It is worth noting that providing a precise computational complexity is challenging due to the involvement of various operations, such as saving samples to the database, retrieving data from the database, and GPU utilization. However, we have provided the maximum number of calls in the response of **Q10**. In practice, for the LLM experiment, for REPP_{N=20, M = 100}, it takes 65 mins in the case that none exists ($\log P < -136$) for the case with $log P = -63.55$, it takes 36 mins to run.
>
>
> **Q9**: Could you elaborate a bit more on the argument why for verification a smaller $N$ suffices
>
> > If we simply want to obtain the verification result rather than an accurate probability of the adversarial event happening, then a smaller $N$ won't affect the result of the approaching of the adversarial event, and a smaller $N$ will help to speed up and it only affects the variance of the estimator.
>
> **Q10**: Why does the number of calls for estimating the probability scale with $\mathcal{O}(\log(1/\mathcal{I}))$
>
> > Recall that $t_{y} = -\log P[Y>y]$ and $\mathcal{I}$ is a special case when $y=0$, hence we got $t_{y=0} = -\log \mathcal{I} = \log(1/\mathcal{I})$, so in the worst case, the total number of calls will be $MN\log(1/ \mathcal{I})$.
>
>
> **Q11**: Minor things
>
> > We thank the reviewer for pointing this out. We have corrected all these small issues. The content regarding the concept of a level was deleted by accident, we have put them at the beginning of Appendix C for now (colored in blue). We will revise the paper to make it more self-contained. We would also like to point out the reviewer to see Figures 5-7 in Appendix C-D to understand the difference between the two.
>
>
> Thanks again for your very long but constructive review. If you find our response satisfactory, please consider raising your score.

---

> > ### Comment · Reviewer_VVLd · 2024-12-02
> >
> > I appreciate the response and improvements made by the authors.
> >
> > I do think the adaptations made to the draft improve the submission. However, even if the authors now mention that they adapted the method from Walter, 2015 in the abstract and introduction; the current presentation of the method needs a major revision which (i) makes clear which concepts / definitions are from Walter and which are new; (ii) what is the actual background from Walter, 2015 necessary to understand the statistical estimation problem for the given ML application; (iii) what is the technical contribution to ML, besides just using the estimator developed by Walter, 2015 as is. Still, nearly the whole methods section and thus technical content of the paper is "just" a repetition of an existing method in a theorem/proof style. Thus, the actual contribution of this work so far seems empirical, for which I would (next to a different presentation), also expect a stronger and more expansive experimental section.
> >
> > I encourage the authors to further pursue their work, but I do think that with the current presentation and contribution this work is not yet ready for ICLR and thus, keep my score.

---

### Official Review · Reviewer_jZjZ · 2024-11-04

**Soundness:** 2
**Presentation:** 1
**Contribution:** 2
**Rating:** 3
**Confidence:** 3

**Summary:**

The paper focuses on the problem of calculating the occurrence of adversarial examples. The problem is modeled as the exponential of the mixture of a Poisson random variable and some potential geometric random variables. A Minimum Variance Unbiased Estimator (MVUE) is developed to accurately estimate the likelihood of encountering adversarial examples. Different from previous works, the proposed method does not require the continuity assumption or the concept of levels. Experiments are conducted on both CV and NLP areas.

**Strengths:**

- Experiments have covered both CV and NLP areas.

**Weaknesses:**

- Motivation is unclear and usefulness is doubtful. The estimated adversarial occurrence seems extremely rare and I wonder how useful this information is. For example, Table 2 shows the log probabilities for adversarial occurrence are lower than -50, implying that the probabilities for adversarial occurrence are basically always zero. How are model developers or users supposed to use these numbers?
- The paper is hard to follow.
- - Preliminaries are not well covered. For example, not requiring the level concept is claimed as one of the main benefits of the proposed method beyond the existing work AMLS. However, neither the AMLS nor the meaning of "level" is introduced. As a result, it's hard to assess the novelty of this paper.
- - Notations are confusing. For example, lowercase $y_0$ is the ground truth label, while uppercase $Y$ is the logit margin and $Y_0$ seems like a random walk state. Another instance is $\mu(x)$, instead of $\mu(x_0)$, denotes the input domain of $x_0$.
- - What is $t_j$ in Equation 12?
- Typos:
- - For equation 2 integral part, I suppose it should be $1_{s(x)\geq0}$?

**Questions:**

Please refer to those in the weaknesses section.

---

> ### Author Response · Authors · 2024-11-28
> **Response to reviewer jZjZ**
>
> We truly appreciate the great comments from the reviewer, and here are our responses to the proposed questions. If you find our response satisfactory, please consider raising your score.
>
> **W1**: Motivation is unclear and usefulness is doubtful. The estimated adversarial occurrence seems extremely rare and I wonder how useful this information is. For example, Table 2 shows the log probabilities for adversarial occurrence are lower than -50, implying that the probabilities for adversarial occurrence are basically always zero. How are model developers or users supposed to use these numbers?
>
> > Actually this example further validates why currently LLMs are popular in people's daily lives, where the low log probabilities of adversarial occurrence illustrate the hardness of a normal user to induce the harmful content from the LLM with a prompt with a random suffix.
> However, for researchers in the adversarial community, a low probability does not represent the non-existence of an adversary, where they can still be induced with advanced adversarial strategy.
>
> **W2**: Preliminaries are not well covered. For example, not requiring the level concept is claimed as one of the main benefits of the proposed method beyond the existing work AMLS. However, neither the AMLS nor the meaning of "level" is introduced. As a result, it's hard to assess the novelty of this paper.
>
> > This part of the content was deleted by accident, we have put it at the beginning of Appendix C for now (colored in blue). We will revise the paper to make it more self-contained.
> We would also like to point out the reviewer to see Figures 5-7 in Appendix C-D to understand the difference between the two.
>
>
> **W3**: Notations are confusing. For example, lowercase $y_0$ is the ground truth label, while uppercase $Y$ is the logit margin and $Y_0$ seems like a random walk state. Another instance is $\mu(x)$ instead of $\mu(x_0)$ denotes the input domain of $x_0$
>
> > We apologize that the notations make you confused but you indeed understand most of them correctly: lowercase $y_0$ is the ground truth label, uppercase $Y$ is the logit margin and $Y_i$ represents the random walk state.
> Regarding $\mu{x}$, given $x_0$ and $x=x_0+\delta$, $\mu(\cdot)$ denotes some distribution, then $\mu(x)$ indeed correctly represnts the distribution of the variable $x$, rather than your mentioned $\mu(x_0)$.
> In other words, $\mu(x)$ denotes the input domain around the original example $x_0$ where adversarial examples may occur.
>
> **W4**: What is $t_j$ in Equation 12?
>
> > In Equation 12, $t_j$ represents the $j$-th target word in the response sentence. The total Equation 12 can be understood as a sum of logit margins among all output targets, it forces the corresponding output to produce the target token at each place, thus producing the whole target sentence eventually.
>
> **W5**: Typos in equation 2 integral part.
>
> > Yes, it was a typo and we thank the reviewer for pointing this out, we will revise this issue.

---

### Official Review · Reviewer_tzGL · 2024-11-05

**Soundness:** 3
**Presentation:** 3
**Contribution:** 2
**Rating:** 5
**Confidence:** 3

**Summary:**

The paper presents a novel statistical method for evaluating the robustness of DNNs through a framework called REPP, which enhances the Adaptive Multi-Level Splitting (AMLS) approach. They treat adversarial instances as rare events and estimates their probability as the exponential of the mixture of a Poisson random variable and some potential geometric random variables.

**Strengths:**

- Motivation for statistical methods looks good- pointing out the limitations and impracticality of deterministic methods effectively.
- The framework demonstrates substantial time efficiency compared to baseline methods.
- It is applicable across a wide range of domains.

**Weaknesses:**

- A bit difficult to interpret Table-1. For certified Acc' which result is better? REPP does not have the best numbers.
- No baseline for results in Table 2.

**Questions:**

- Could the results be compared with Randomized Smoothing (another statistical verification technique) or other certified accuracy methods, like direct certification via IBP, on commonly used datasets for adversarial robustness, such as CIFAR-10 or Tiny IMN? (or a justification on why that is not needed)

---

> ### Author Response · Authors · 2024-11-28
> **Response to reviewer tzGL**
>
> We truly appreciate the great comments from the reviewer, and here are our responses to the proposed questions. If you find our response satisfactory, please consider raising your score.
>
> **W1.** A bit difficult to interpret Table-1. For certified Acc' which result is better? REPP does not have the best numbers.
>
> > For certified Acc, it is true that higher is better, but please note that this is evaluated with a specified tolerance error $\tau$.
> GeoRobust is a deterministic verification technique that can be seen as a baseline with $\tau=0$, and its attacking performance should be a valid upper bound.
> PRoA is a statistical verification technique that relies on independent sampling with concentration inequality.
> In Table 1 we show that under the same confidence threshold, REPP can achieve more meaningful results: $PRoA_{
> \tau=0.02}$ and $PRoA_{\tau=0.01}$ show significant false negatives as their certified accuracies are greater than the attack accuracy in several cases, but with smaller tolerance error $\tau=0.01$, its certified accuracy drops into 0%, providing nothing useful for the evaluation (this also happens to other concentration inequality based methods); Instead, with far more small tolerance error $\tau=10^{-15}$, REPP still offer a valid evaluation for robustness.
>
> >We will refine our content to ensure greater clarity.
>
> **W2.** No baseline for results in Table 2.
>
> > There are indeed no other baselines that can show some relevant statistics like us, we try random sampling 10^10 times but get 0 valid adversarial suffix showing up, which further validates the event's rarity.
>
>
> **Q1.** Could the results be compared with Randomized Smoothing (another statistical verification technique) or other certified accuracy methods, like direct certification via IBP, on commonly used datasets for adversarial robustness, such as CIFAR-10 or Tiny IMN? (or a justification on why that is not needed)
>
> > Although the method we proposed falls into statistical verification, the definition are entirely different from Randomized Smoothing.
> It is noted that Randomized Smoothing can be seen as a defense approach, and it aims to construct a smoothed classifier that is robust to input perturbations and provides certified adversarial robustness on the **smoothed classifier**.
> However, our method, REPP directly estimates the probability of encountering adversarial examples on the **base classifier**, therefore they can not be compared directly in this case.
> One possible comparison we can see is to compute the probability of the adversarial example of the smoother classifier given the certified radius by RS, to see whether both probabilities match.
> However, this may be a bit out of the scope of this work, but we are willing to put this comparison in the future study.
>
> > Regarding other certified accuracy methods, as shown in Table 1, we compared our method with GeoRobust as a baseline, a deterministic verification based on the Lischitz optimization search.
> The main results we are comparing are other statistical verification approaches, showing that methods based on concentration inequality may lose their effectiveness when the existence of adversarial examples is really rare.
> We do not aim to claim a statistical approach can beat deterministic verification, instead, it can provide more information and some confidence in the model deployed when deterministic methods are infeasible.

---

> > ### Comment · Reviewer_tzGL · 2024-11-30
> >
> > Thanks for your rebuttal, based on the results and baselines, I will keep my score.

---

### Note · Authors · 2025-01-22

**Comment:**

We sincerely thank the reviewers for their invaluable feedback and time. We will incorporate the provided comments to improve our next submission.

**Withdrawal Confirmation:**

I have read and agree with the venue's withdrawal policy on behalf of myself and my co-authors.